# Effect of Dietary Lactose Supplementation on Growth Performance and Intestinal Epithelium Functions in Weaned Pigs Challenged by Rotavirus

**DOI:** 10.3390/ani12182336

**Published:** 2022-09-08

**Authors:** Wei Yu, Xuechun Xiao, Daiwen Chen, Bing Yu, Jun He, Ping Zheng, Jie Yu, Junqiu Luo, Yuheng Luo, Hui Yan, Xuewu Yi, Jianping Wang, Huifen Wang, Quyuan Wang, Xiangbing Mao

**Affiliations:** 1Key Laboratory of Animal Disease-Resistant Nutrition and Feed of China Ministry of Agriculture and Rural Affairs, Institute of Animal Nutrition, Sichuan Agricultural University, Chengdu 611130, China; 2College of Life Sciences, Leshan Normal University, Leshan 614000, China

**Keywords:** lactose, rotavirus, growth performance, nutrient utilization, gut health

## Abstract

**Simple Summary:**

Lactose is a kind of carbohydrate that exists in mammal milk. It has some physiological functions, such as providing energy, regulating gut microbiota, and affecting immunity. Rotavirus (RV) is the main pathogen that induces severe diarrhea in piglets, which impairs their growth and development. In this study, we investigated whether different levels (4% and 6%) of dietary lactose supplementation alleviates RV-induced diarrhea in weaned piglets. The results showed that lactose administration relieved the negative effect of RV on growth, which was derived from the improvement of nutrient utilization, gut barrier function, and immunity. Moreover, supplementing 6% lactose in the diets had a tendency to alleviate diarrhea in RV-infected piglets. Thus, we suggest that the diet of weaned piglets should be supplemented with more than 4% lactose (especially in the early period of weaning) if the cost of feed can be afforded.

**Abstract:**

The purpose of this study was to investigate whether dietary lactose supplementation relieves rotavirus (RV)-induced diarrhea and gut dysfunction. Thirty-six crossbred weaned piglets were randomly allocated into three groups and fed diets containing 0, 4%, and 6% lactose for 20 days. On Day 15, half of the piglets in each group were orally infused with RV. RV infection impaired growth performance; induced severe diarrhea; decreased serum D-xylose concentration and morphology and sIgA level of jejunal mucosa; downregulated MUC1, MUC2, occludin, Bcl-2, IL-4, pBD3, pBD2, and pBD1 mRNA expression of jejunal mucosa and/or mesenteric lymph nodes; upregulated Bax, caspase-3, IL-2, IFN-γ, and IFN-β mRNA expression of jejunal mucosa and/or mesenteric lymph nodes; and damaged microbiota and metabolites of cecal digesta in weaned piglets (*p* < 0.05). Dietary lactose supplementation improved nutrient digestibility and growth performance and relieved the negative influence of RV challenge on intestinal barrier function, mRNA expression of cytokines, and host defense peptides of jejunal mucosa and/or mesenteric lymph nodes in weaned piglets *(p* < 0.05). Dietary administration of 6% lactose tended to relieve diarrhea *(p* = 0.07). These results suggest that lactose in feed increases growth performance and has a tendency to alleviate RV-induced diarrhea, derived from the improvement of nutrient utilization, gut barrier function, and immunity.

## 1. Introduction

Rotavirus (RV) is the main pathogen that induces severe diarrhea in infants, children, and young animals [1,2]. RV infection will damage the intestinal mucosa of the host, cause the disorder of nutrient absorption and water electrolytes, and finally lead to host diarrhea [3]. As a double-stranded RNA virus, after entering the small intestine, RV will invade the epithelial cells and induce an inflammatory response, immune dysfunction, and oxidative stress, which can be the main causes of RV damaging gut barrier function and inducing diarrhea [4,5,6,7,8]. The comprehensive infection rate of piglets (1–4 weeks old) is over 80%, and the mortality rate is as high as 20%. Moreover, infected breeding sows can become an invisible source of infection and further result in the continuous occurrence of RV infection in the local area [9].

Lactose, known as a kind of carbohydrate that exists in mammal milk, is resolved into glucose and galactose by lactase. This can supply energy to infants, children, and young animals [10]. In the large intestine, unabsorbed lactose may be utilized and fermented into short-chain fatty acids and lactic acid, and then the pH value of the enteric internal environment will be decreased, which will be beneficial for gut microflora [11,12,13]. In particular, the activity of lactase in the gut will decrease with the increasing age of piglets. This increases the level of lactose in the large intestine, where its functions are mainly embodied via itself or short-chain fatty acids and lactic acid after fermentation by gut microflora. [11,12,13]. Recently, research has reported that lactose administration could stimulate the generation of host defense peptides (HDPs) in human intestinal cells and phagocytes [10]. Our previous study also showed that lactose treatment upregulated HDPs (such as porcine β-defensins (pBD), protegrin 1-5 (PG1-5)), mucins, tight-junction-related proteins, and antivirus cytokines (e.g., interferons (IFN)) in IPEC-J2 cells and/or 3D4/31 cells (data not published). Lactose has a potential role in maintaining gut health.

Based on the physiological functions of lactose, it is possible that dietary lactose supplementation could improve immunity and gut health and attenuate RV-induced diarrhea. Whey has already been used in the feed of young animals (especially piglets), the main component of which is lactose [14]. According to the addition percentage of whey, generally, the content of lactose is approximately 4% in piglet feed. It is unknown whether increasing the content of lactose is more effective than 4% lactose. Therefore, the purpose of this research was to (1) test the hypothesis that supplementing different lactose levels in the diets relieves RV-induced diarrhea and improves growth performance and (2) analyze the potential change in gut health in this process.

## 2. Materials and Methods

### 2.1. Ethical Approval

The experimental protocol was approved by the Animal Care Advisory Committee of Sichuan Agricultural University (Chengdu 611130, China). All experimental protocols were approved by the Institutional Animal Care and Use Committee of the Laboratory Animal Center at Sichuan Agricultural University on 24 June 2019 (SICAU-2019-026). Animal testing was enforced at the Experimental Farm of Sichuan Agricultural University.

### 2.2. Animals and Diets

A total of 36 Duroc × Landrace × Yorkshire (DLY) 21-day-old weaned piglets (barrow) with similar initial body weight (BW, 6.30 ± 0.10 kg) from 12 litters were individually fed in metabolic cages (1.50 × 0.70 × 1.00 m^3^). All piglets were fed powders 4 times daily at 08:00, 12:00, 16:00, and 20:00 and received water ad libitum. The quantity of the feeding diet was based on the principle that there was a little remaining feed in the trough after feeding, and the remaining feed in the trough was cleaned, weighed, and recorded at 10:00 p.m. every day.

The basal diet was formulated to meet or exceed the nutrient recommendation for 7–11 kg pigs in NRC [15], of which the composition and nutrient contents are exhibited in Table 1. The experimental diets were formed by supplementing 0, 4%, and 6% lactose in the basal diet by replacing corn starch. The lactose was purchased from Baiying Biotechnology Co., Ltd., Jiangxi, China, with a purity of 99%.

### 2.3. Experimental Design and Sample Collection

After 3 days of orientation, on the basis of initial body weight and litter origin, all piglets were randomly allocated into 3 groups (*n* = 12) and were fed with the experimental diets containing 0, 4%, and 6% lactose for 20 days. During Days 11–14, nutrient digestion was evaluated as described previously [16].

RV preparation and virus titer measurement (tissue culture infective dose 50 (TCID_50_) value) were carried out as described previously [17].

On Day 15, all piglets were gavaged, and the specific gavage methods were carried out as described previously [18]. Briefly, all piglets (39 days old) were administered 5 mL of sterile 100 mmol/L NaHCO_3_ solution via oral gavage. After 15 min, half of the piglets in each group were orally infused with 5 mL (10^6^ TCID_50_/mL) of rotavirus (RV+), while the other piglets orally received 5 mL of sterile essential medium (RV−). Then, the diarrhea status of all piglets was observed and recorded each day. The scoring method of fecal consistency was carried out as described previously [18]. In brief, normal, 0; pasty, 1; semiliquid, 2; liquid, 3. A fecal consistency score of ≥2 was considered diarrhea. Feed intake was recorded on each day of the experiment, and animals were weighed on Days 1, 15, and 21. The growth performance indexes (including average daily gain (ADG), average daily feed intake (ADFI), and feed/weight gain ratio (F/G)) of all piglets were calculated from the above-recorded data.

On Day 21, following 12 h of starvation, the piglets were weighed and infused with sterile 10 mg/mL D-xylose solution (1 mL/kg of body weight). The D-xylose was purchased from Haoboyou Biotechnology Co., Ltd., Chengdu, China, with a purity of 99%. One hour later, the blood sample of all piglets was gathered from the jugular vein, centrifuged at 3500× *g* for 10 min, and serum was collected. Then, after refeeding for 2 h, all piglets were euthanized via intracardiac injection with sodium pentobarbital (50 mg/kg of body weight) and exsanguinated. The whole intestine was removed. The jejunum was quickly separated and flushed with ice-cold saline solution. Lymph nodes were immediately excised. The sample of jejunal mucosa was collected by scraping the gut wall with the glass microscope slide. About 2 cm jejunal segments were fixed in 4% paraformaldehyde. About 3 g of cecal digesta was gathered in sterile tubes, and the other cecal digesta were used to measure the pH value using a pH meter (PHS-3C). The samples of lymph nodes, jejunal mucosa, and cecal digesta were immediately frozen in liquid nitrogen and stored at −80 °C.

### 2.4. Nutrient Digestibility

In the digestion trial, ash insoluble in hydrochloric acid was the nonabsorbable digestion marker. Dry matter (DM), crude protein (CP), gross energy (GE), ether extract (EE), ash, and ash insoluble in hydrochloric acid in feeds and feces were determined according to the previous description [16]. The digestibility of these nutrients was calculated as previously described [18].

### 2.5. Serum Urea Nitrogen (UN) and D-Xylose Levels

Serum UN (C013-2-1) and D-xylose (A035-1-1) concentrations were analyzed by using the kit (Nanjing Jiancheng Bioengineering Institute, Nanjing, China) according to the manufacturer’s instructions.

### 2.6. Rotavirus Nonstructural Protein 4 (NSP4), Secreting Immunoglobulin A (sIgA), and Antioxidant Capacity in Jejunal Mucosa

Approximately 100 mg of jejunal mucosa was added into ice-cold PBS, shattered at 4 °C, and then centrifuged at 5000× *g* for 15 min at 4 °C. The supernatants were collected and were used to measure the related index.

The NSP4 concentration in jejunal mucosa was measured with an ELISA kit (TSZ ELISA, Framingham, MA, USA). The sIgA concentration in jejunal mucosa was detected with ELISA kits from Nuoyuan Co. Ltd. (Shanghai, China). Malondialdehyde (MDA) level, total antioxidant capacity (T-AOC), and total protein level from jejunal mucosa were detected by the kit of Nanjing Jiancheng Bioengineering Institute (Nanjing, China).

### 2.7. Morphology of Jejunal Mucosa

The morphology in jejunal mucosa was determined as described previously [4]. In brief, following fixing in 4% paraformaldehyde, the jejunal segment was embedded in paraffin. Then, consecutive sections (5 µm) were stained with hematoxylin–eosin. In each sample, a total of 10 intact villi and crypts were randomly selected, and villus height and crypt depth were measured at 40× magnification with an Olympus CK 40 microscope.

### 2.8. mRNA Expression of Some Genes in Jejunal Mucosa and/or Mesenteric Lymph Nodes

Total RNA in jejunal mucosa and mesenteric lymph nodes was extracted with TRIZOL reagent (TaKaRa Biotechnology (Dalian) Co., Ltd., Dalian, China) according to the manufacturer’s instructions. RNA concentrations were determined with DU 640 UV spectrophotometer detection (Beckman Coulter Inc., Fullerton, CA, USA). The OD_260_/OD_280_ ratio was 1.8 to 2.0. RNA quality of samples was evaluated with 1% agarose gel electrophoresis. Then, in all samples, the cDNA was synthesized by using RT Reagents (TaKaRa Biotechnology (Dalian) Co., Ltd., Dalian, China) according to the manufacturer’s instructions. All genes’ primers of this study are listed in Table 2 and obtained from TaKaRa Biotechnology (Dalian) Co., Ltd. (Dalian, China). The mRNA expression of mucin 1 (MUC1), MUC2, zonula occludens-1 (ZO-1), occludin, B-cell lymphoma/leukemia-2-associated X protein (Bax), B-cell lymphoma/leukemia-2 (Bcl-2), interleukin 2 (IL-2), IL-4, interferon γ (IFN-γ), IFN-β, porcine β-defensin 1 (pBD1), pBD2, pBD3, and β-actin in jejunal mucosa and/or mesenteric lymph nodes was determined by real-time quantitative PCR with SYBR Premix Ex Taq reagents (TaKaRa Biotechnology (Dalian) Co., Ltd., Dalian, China) and CFX-96 Real-Time PCR Detection System (Bio-Rad Laboratories, Richmond, CA, USA) as described previously [18]. To verify the variance in the amount of RNA input of reaction, β-actin was utilized as the reference gene. Then, the relative mRNA expression was obtained by using the previous method [19].

### 2.9. Populations of Some Bacteria in the Cecal Digesta

Bacterial DNA of cecal digesta was extracted with the Stool DNA Kit (Omega BioTek, Doraville, CA, USA) according to the manufacturer’s instructions. The bacterial real-time quantitative PCR and related result transformation were executed as described previously [4]. All primers and probes, listed in Table 3, were purchased from TaKaRa Biotechnology (Dalian) Co., Ltd. (Dalian, China).

### 2.10. Short-Chain Fatty Acids (SCFAs) in the Cecal Digesta

SCFAs (including acetic acid, propionic acid, isobutyric acid, butyric acid, isovaleric acid, and valeric acid) levels were analyzed with the Varian CP-3800 gas chromatograph (Agilent Technologies, Santa Clara, CA, USA) as described previously [4].

### 2.11. Statistical Analysis

The index of growth and nutrient digestibility of piglets before RV challenge was analyzed by using one-way ANOVA, followed by Duncan’s multiple range test. The data on diarrhea rate of piglets were analyzed through the chi-square test. The other data were analyzed as a 2 × 3 factorial with the general linear model procedures of the Statistical Analysis Package. The model factors were involved in dietary lactose supplementation, rotavirus challenge, and their interaction. These analyses were executed with SPSS (Version 21.0, IBM, Armonk, NY, USA). All data were embodied as means and their SEs. *p* < 0.05 was considered statistically significant, whereas *p* < 0.10 was considered a statistical tendency.

## 3. Results

### 3.1. Growth Performance, Serum UN, and Nutrient Digestibility in Weaned Piglets

In the first two weeks of experimental duration, dietary lactose supplementation had a disposition to increase ADG (*p* = 0.09, Table 4), significantly decrease F/G (*p* < 0.05, Table 4), and significantly increase the digestibility of CP, GE, and ash (*p* < 0.05, Figure 1) in weaned piglets. Compared with piglets fed by the basal diet, piglets fed the diet supplemented with 6% lactose had lower F/G and higher digestibility of GE (*p* < 0.05), but piglets fed the diet supplemented with 4% lactose had no significant effects on F/G and GE digestibility (Table 4 and Figure 1). From Days 15 to 20, RV infection significantly enhanced F/G and serum UN (*p* < 0.05), and supplementing lactose in the diets significantly reduced F/G and serum UN *(p* < 0.05) in weaned piglets (Table 4). In RV-infused piglets, supplementing lactose in the diets significantly decreased serum UN (*p* < 0.05), but there was no significant difference between piglets fed diets supplementing 4% and 6% lactose (Table 4).

### 3.2. Diarrhea Status and NSP4 Levels and Antioxidant Capacity of Jejunal Mucosa in Weaned Piglets

As shown in Table 5, the essential medium by oral gavage did not lead to diarrhea, but RV infusion significantly induced diarrhea in weaned piglets (*p* < 0.05). Rotavirus infection also significantly enhanced NSP4 and MDA concentrations and inhibited T-AOC in the jejunal mucosa of weaned piglets (*p* < 0.05, Table 6). The effect of adding lactose to the diet on the diarrhea rate depends on the amount of lactose. Supplementing 4% lactose in the diet had no effect on the diarrhea rate, while the administration of 6% lactose had a tendency to alleviate diarrhea caused by RV (*p* = 0.07, Table 5). In addition, supplementing lactose in the diets significantly decreased the NSP4 level and alleviated the increasing NSP4 level induced by RV infusion in the jejunal mucosa of weaned piglets (*p* < 0.05, Table 6). However, lactose administration did not significantly affect antioxidant capacity in the jejunal mucosa of weaned piglets (Table 6).

### 3.3. Serum D-Xylose Concentration, and Morphology of Jejunal Mucosa in Weaned Piglets

Following RV challenge, serum D-xylose concentration was significantly decreased (*p* < 0.05, Table 7), and the villus height/crypt depth of jejunal mucosa was also significantly reduced (*p* < 0.05, Table 8) in weaned piglets. However, supplementing lactose in the diets significantly increased serum D-xylose concentration and villus height/crypt depth of jejunal mucosa *(p* < 0.05, Table 7 and Table 8). In contrast with the piglets fed the diet containing 4% lactose, there was a more efficient serum D-xylose concentration in piglets fed the diet containing 6% lactose (Table 7).

### 3.4. mRNA Expression of Barrier-Related and Apoptosis-Related Genes in the Jejunal Mucosa of Weaned Piglets

In the jejunal mucosa of weaned piglets, the mRNA expression of MUC1, MUC2, occludin, and Bcl-2 was significantly downregulated by RV infection, and the mRNA expression of Bax and caspase-3 was significantly upregulated by RV infection (*p* < 0.05, Table 9). Supplementing lactose in the diets significantly stimulated the mRNA expression of MUC1, MUC2, ZO-1, occludin, and Bcl-2, inhibited the mRNA expression of Bax and caspase-3, and then relieved the effect of RV infection on the mRNA expression of MUC1, MUC2, ZO-1, occludin, Bax, and Bcl-2 in the jejunal mucosa of weaned piglets (*p* < 0.05, Table 9). Furthermore, in the jejunal mucosa of RV-infected piglets, the diet containing 6% lactose was more efficient in improving the mRNA expression of MUC1, MUC2, ZO-1, and occludin than the diet containing 4% lactose (Table 9).

### 3.5. mRNA Expression of Cytokines and Host Defense Peptides, and sIgA Level in the Jejunal Mucosa and/or Mesenteric Lymph Nodes of Weaned Piglets

Following RV challenge, the sIgA level of jejunal mucosa was significantly decreased (*p* < 0.05, Table 6); the IL-2, IFN-γ, and IFN-β mRNA expression of jejunal mucosa and mesenteric lymph nodes was significantly stimulated (*p* < 0.05, Table 10); and the IL-4, pBD1, pBD2, and pBD3 mRNA expression of jejunal mucosa and/or mesenteric lymph nodes was significantly downregulated (*p* < 0.05, Table 10) in weaned piglets. Supplementing lactose in the diets significantly enhanced the sIgA level of jejunal mucosa; upregulated the IL-4, IFN-γ, IFN-β, pBD1, pBD2, and pBD3, mRNA expression of jejunal mucosa and/or mesenteric lymph nodes; and reduced the IL-2 mRNA expression of jejunal mucosa and mesenteric lymph nodes in weaned piglets (*p* < 0.05, Table 6 and Table 10). In RV-infected piglets, lactose administration significantly relieved the negative influence of rotavirus infection on the sIgA level of jejunal mucosa; the IL-2, IL-4, pBD1, and pBD3 mRNA expression of jejunal mucosa and/or mesenteric lymph nodes; and further promoted the IFN-γ and IFN-β mRNA expression of jejunal mucosa and/or mesenteric lymph nodes (*p* < 0.05, Table 6 and Table 10). Additionally, compared with the diet containing 4% lactose, the diet containing 6% lactose was more efficient in regulating sIgA concentration and the mRNA expression of IFN-γ, IFN-β, pBD1, and pBD3 in the jejunal mucosa of RV-infected piglets (Table 6 and Table 10).

### 3.6. Population of Some Bacteria, SCFA Levels and pH Value in the Cecal Chyme of Weaned Piglets

RV challenge significantly reduced the number of *Bacillus* and *Bifidobacterium*, significantly increased the number of *Escherichia coli* (*E. coli*), and significantly reduced the consistency of acetic acid, isobutyric acid, and total SCFA in the cecal chyme of weaned piglets (*p* < 0.05, Table 11 and Table 12). Supplementing lactose in the diets significantly enhanced the population of *Lactobacillus* (*p* < 0.05); significantly upregulated the concentrations of acetic acid, propionic acid, and total SCFA (*p* < 0.05); and tended to downregulate the pH value (*p* = 0.06) in the cecal chyme of weaned piglets (Table 11 and Table 12). Dietary lactose supplementation significantly relieved the effect of RV challenge on the pH value in the cecal chyme of weaned piglets (*p* < 0.05, Table 12). Moreover, in the RV-infected piglets, compared with the diet containing 4% lactose, the diet containing 6% lactose was more efficient in increasing the levels of propionic acid and total SCFA in the cecal chyme (Table 12).

## 4. Discussion

Pathogen-induced diarrhea is the main cause that negatively affects health and/or growth in humans and animals, especially infants, children, and young animals [3]. Rotavirus (RV) is one of the major pathogens that induce severe diarrhea in young humans and animals by damaging gut health [1,2,20]. Our previous studies have reported that RV infection causes diarrhea, a decrease in growth performance, dysfunction of the gut barrier, inflammation, and changes in the gut bacteria of weaned pigs [4,5,6,17,18,20,21]), similar to the results of this study. Besides these, we also found that RV challenge increased RV NSP4 levels of jejunal mucosa and decreased serum D-xylose consistency in weaned piglets. Thus, these results illustrated that RV infection in weaned piglets was successfully implemented.

Previous research has shown that supplementing lactose in diets might increase the growth performance of weaned piglets and nursery pigs, and the influence of lactose administration on growth performance is dose-dependent [22,23,24]. The present research achieved similar results, showing that supplementing lactose in the diets improved the F/G of weaned piglets and tended to improve the ADG of weaned piglets and the F/G of RV-infected piglets. These results also embodied the better effect of high-dose lactose. It is well known that the digestion, absorption, and utilization of nutrients play a vital role in the growth of humans and animals. In this research, nutrient digestibility and serum UN in weaned piglets were measured, and we discovered that lactose administration raised the digestibility of CP, GE, and ash and decreased serum UN, known as a potential marker of nitrogen (especially protein and amino acid) utilization, in weaned piglets. These illustrated that supplementing lactose in the diets improving the growth performance of piglets was associated with an increase in nutrient digestion, absorption, and utilization. Moreover, our results showed that lactose administration increasing growth performance on Days 1 to 14 was more efficient than that on Days 15 to 20. Therefore, lactose should be supplemented in the feed of piglets during the initial stage of weaning.

Weaning is often coupled with intestinal morphology damage, which has been widely applied to evaluate the intestinal health or functions of piglets [25]. Gut epithelial integrity is one of the important causes that maintain digestion and absorption functions, and the surface area of villi in the intestinal mucosa is related to nutrient digestion and absorption. In the current research, lactose administration did not affect villus height and crypt depth but, to some extent, improved villus height/crypt depth in the jejunal mucosa of normal and/or RV-infected piglets. Despite this, we also found that dietary lactose supplementation could increase serum D-xylose levels in piglets, which could reflect the positive effect of lactose on gut absorption function and epithelial integrity.

NSP4 is a nonstructural protein of RV, which is a vital factor that infects gut epithelial cells and induces severe diarrhea in humans and animals. NSP4 is often regarded as an important marker of RV infection [26]. The results of our research showed that RV challenge elevated NSP4 concentrations, but lactose administration decreased NSP4 concentrations in the jejunal mucosa of weaned piglets. Thus, it is possible that supplementing lactose in the diets inhibited RV invasion and infection in the intestine of piglets.

Intestinal barrier function is vital for protecting gut health from pathogen invasion [27]. Epithelial tight junctions (TJs) are a critical physical barrier against the permeation of many molecules (such as pathogens, toxins, and antigens) from the luminal environment into the mucosal tissues and circulatory system [28]. Lactose administration potentially improved the intestinal barrier function in RV-infected piglets. Thus, except for mucosal epithelial integrity (such as mucosal morphology and serum D-xylose levels), we analyzed the gut-barrier-related indexes. The intercellular junctions between epithelial cells of the mucosa are mainly associated with transmembrane and no membrane proteins, such as ZO-1 and occludin [29]. The main component of the mucus gel layer is mucins generated by the goblet and epithelial cells of the mucosa [30]. In addition, immunity is also pivotal to preventing pathogen (especially virus) invasion in the gut, which includes some immune macromolecules, including sIgA, interferons, and host-defense peptides [31,32,33]. In our research, supplementing lactose in the diets alleviated the influence of RV infection on sIgA level and MUC1, MUC2, ZO-1, occludin, pBD1, and pBD3 mRNA expression of the jejunal mucosa, and further promoted IFN-γ and IFN-β mRNA expression of jejunal mucosa and/or mesenteric lymph nodes in weaned piglets. These illustrated that supplementing lactose in the diets protecting the gut from RV infection was related to the improvement of barrier function.

In the process of maintaining gut barrier function, the activity of mucosal cells plays an important role. Bax and Bcl-2 are apoptosis-related proteins [34]. The results of our study showed that supplementing lactose in the diets relieved the negative influence of RV infection on the mRNA expression of Bax and Bcl-2 in the jejunal mucosa of weaned piglets. Impairing intracellular redox status and increasing inflammation are two important ways to increase RV-induced apoptosis of gut mucosal cells (especially in the late period of RV infection) [5]. In this study, supplementing lactose in the diets did not regulate the decreasing antioxidant capacity of the jejunal mucosa, but alleviated the increasing proinflammatory cytokine (IL-2) and decreasing anti-inflammatory cytokine (IL-4) mRNA expression of jejunal mucosa and/or mesenteric lymph nodes in RV-infected piglets. Thus, lactose administration improving the gut barrier function of RV-infected piglets could result from the regulated apoptosis of mucosal cells possibly via the downregulated inflammation.

Gut microbiota and their metabolites (including SCFAs) also play a critical role in gut barrier function [35,36]. Previous researches have reported that lactose administration can increase the number of beneficial bacteria in the intestine and SCFA levels in humans and animals [11,12,13], which is similar to the results of our research. It was possible that supplementing high-dose lactose in the diets attenuating diarrhea induced by RV infection, at least partially, should be in contact with the improvement of gut microbiota in piglets.

In addition, we also found that high-dose lactose administration had a more significant effect on improving growth performance (especially in the first two weeks) and reducing RV-induced diarrhea. After analyzing all indexes, this could be due to the fact that the improvement of nutrient digestibility (such as GE) and gut barrier functions (including the mRNA expression of mucins, tight junctions, and immunity) in piglets with a diet supplemented with 6% lactose is more efficient than that in piglets with a diet supplemented with 4% lactose.

## 5. Conclusions

RV infection could impair growth performance and induce diarrhea and gut dysfunction in weaned piglets. Dietary lactose supplementation attenuated these negative effects of RV challenge on the growth performance of piglets, which was derived from the improvement of nutrient digestibility, gut barrier function, and immunity. According to our results, the administration of 6% lactose in the diet is more effective than 4% lactose. Thus, we suggest that the diet of weaned piglets should be supplemented with more than 4% lactose in the general feed (especially in the early period) if the cost of feeds can be afforded.

## Figures and Tables

**Figure 1 animals-12-02336-f001:**
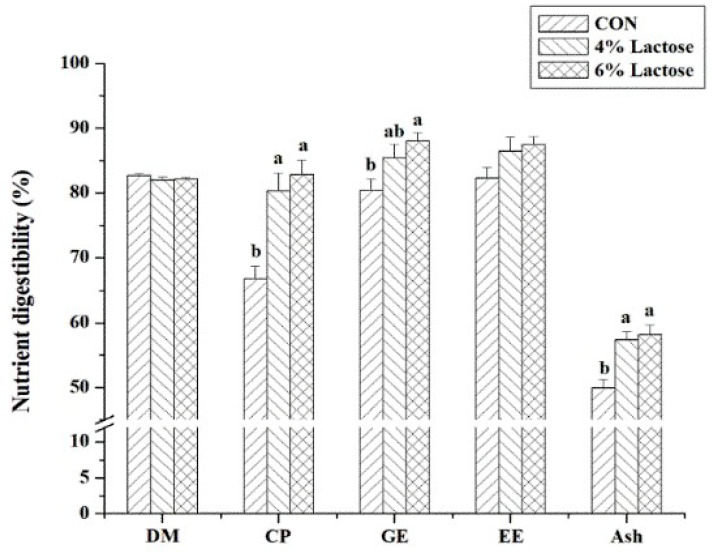
Effect of dietary lactose supplementation on nutrient digestibility in weaned piglets. Values are means ± SEs; *n* = 12. Values with different letters are significantly different (*p* < 0.05). CON, basal diet; 4% Lactose, diet supplemented with 4% lactose; 6% Lactose, diet supplemented with 6% lactose; DM = dry matter; CP = crude protein; GE = gross energy; EE = ether extract.

**Table 1 animals-12-02336-t001:** The composition and nutrient contents of experimental diets (%).

Ingredients	Content		
	0% Lactose	4% Lactose	6% Lactose
Extruded corn	30.00	30.00	30.00
Corn	21.00	21.00	21.00
Soybean meal	16.80	16.80	16.80
Corn starch	10.53	6.53	4.53
Extruded Soybean	7.50	7.50	7.50
Fish protein concentrate	4.00	4.00	4.00
Soybean protein concentrate	4.00	4.00	4.00
Sucrose	2.00	2.00	2.00
Soybean oil	1.60	1.60	1.60
Dicalcium phosphate	0.74	0.74	0.74
Limestone	0.70	0.70	0.70
L-Lysine HCl (78%)	0.31	0.31	0.31
Sodium chloride	0.20	0.20	0.20
Chloride choline (50%)	0.15	0.15	0.15
DL-Methionine (98.5%)	0.08	0.08	0.08
L-Threonine (98.5%)	0.03	0.03	0.03
L-Tryptophan (98%)	0.01	0.01	0.01
Vitamin premix ^1^	0.05	0.05	0.05
Mineral premix ^2^	0.30	0.30	0.30
Lacoste	0.00	4.00	6.00
Total	100.00	100.00	100.00
Nutrition content ^3^			
Digestible energy (Mcal/kg)	3.56	3.55	3.54
Crude protein	19.85	19.83	19.83
Calcium	0.84	0.84	0.84
Phosphorus	0.60	0.60	0.60
Lysine	1.35	1.35	1.35
Threonine	0.79	0.79	0.79
Met + Cys	0.68	0.68	0.68
Methionine	0.40	0.40	0.40
Tryptophan	0.23	0.23	0.23

^1^ The vitamin premix provides per kg of diet: vitamin A, 9000 IU; vitamin D_3_, 3000 IU; vitamin E, 20.0 IU; vitamin K_3_, 3.0 mg; vitamin B_1_, 1.5 mg; vitamin B_2_, 4.0 mg; vitamin B_6_, 3.0 mg; vitamin B_12_, 0.2 mg; nicotinic acid, 30.0 mg; D-pantothenic acid, 15.0 mg; folic acid, 0.75 mg; biotin, 0.1 mg. ^2^ The mineral premix provides per kg of diet: Fe, 100 mg; Zn, 100 mg; Cu, 6 mg; Mn, 4 mg; Se, 0.3 mg; I, 0.14 mg. ^3^ Nutrient contents were the calculated values.

**Table 2 animals-12-02336-t002:** Primer sequences used for real-time PCR.

Genes	Nucleotide Sequences 5′-3′
MUC1	F: GTGCCGCTGCCCACAACCTG
	R: AGCCGGGTACCCCAGACCCA
MUC2	F: GGTCATGCTGGAGCTGGACAGT
	R: TGCCTCCTCGGGGTCGTCAC
ZO-1	F: CTGAGGGAATTGGGCAGGAA
	R: TCACCAAAGGACTCAGCAGG
occludin	F: CTACTCGTCCAACGGGAAAG
	R: ACGCCTCCAAGTTACCACTG
Bcl-2	F: TGCCTTTGTGGAGCTGTATG
	R: GCCCGTGGACTTCACTTATG
Bax	F: AAGCGCATTGGAGATGAACT
	R: TGCCGTCAGCAAACATTTC
Caspase-3	F: GGGATTGAGACGGACAGTGG
	R: TGAACCAGGATCCGTCCTTTG
IFN-β	F: CGATACCAACAAAGGAGCAG
	R: GGTTTCATTCCAGCCAGT
pBD1	F: TCCTTGTATTCCTCCTCA
	R: CAAATCCTTCACCGTCTACCA
pBD2	F: TGTCTGCCTCCTCTCTTCC
	R: AACAGGTCCCTTCAATCCTG
pBD3	F: CCTTCTCTTTGCCTTGCTCTT
	R: GCCACTCACAGAACAGCTACC
IL-2	F: ACCAGGCCATTCAAAGGAGC
	R: CGAAGTCATTCAGTTTCCCAGAG
IL-4	F: CAGCGAGAAAGAACTCGTGC
	R: GGTTTCCTTCTCCGTCGTGT
IFN-γ	F: TGGTAGCTTCTGGGAACTGAATG
	R: AGGCTTTCGCGCTGGATCTGC
β-actin	F: TCTGGCACCACACCTTCT
	R: TGATCTGGGTCATCTTCTCAC

**Table 3 animals-12-02336-t003:** Primer and probe sequences used for real-time PCR.

Bacteria	Nucleotide Sequences 5′-3′
*E. coli*	F: CATGCCGCGTGTATGAAGAA
	R: CGGGTAACGTCAATGAGCAAA
	P: AGGTATTAACTTTACTCCCTTCCTC
*Bacillus*	F: GCAACGAGCGCAACCCTTGA
	R: TCATCCCCACCTTCCTCCGGT
	P: CGGTTTGTCACCGGCAGTCACCT
*Bifidobacterium*	F: CGCGTCCGGTGTGAAAG
	R: CTTCCCGATATCTACACATTCCA
	P: ATTCCACCGTTACACCGGGAA
*Lactobacillus*	F: GAGGCAGCAGTAGGGAATCTTC
	R: CAACAGTTACTCTGACACCCGTTCTTC
	P: AAGAAGGGTTTCGGCTCTAAAACTCTGTT
Total bacteria	F: ACTCCTACGGGAGGCAGCAG
	R: ATTACCGCGGCTGCTGG

**Table 4 animals-12-02336-t004:** Effect of dietary lactose supplementation and/or RV challenge on growth performance and serum UN concentration in weaned pigs ^1^.

	RV−			RV+			*p* Value		
	CON	4% Lactose	6% Lactose	CON	4% Lactose	6% Lactose	Lactose	RV	Lactose × RV
1-14,d									
ADFI,g	278.36 ± 36.61	272.47 ± 32.87	312.02 ± 23.69				0.64		
ADG,g	148.43 ± 22.45	158.14 ± 23.25	210.86 ± 16.49				0.09		
F/G	1.99 ± 0.13 ^a^	1.81 ± 0.10 ^a,b^	1.48 ± 0.03 ^b^				<0.05		
15–20,d									
ADFI,g	513.94 ± 72.40	486.19 ± 12.23	492.56 ± 55.99	534.56 ± 99.55	460.63 ± 94.70	479.38 ± 52.08	0.92	0.76	0.95
ADG,g	358.75 ± 42.49	392.50 ± 21.07	434.25 ± 66.69	302.50 ± 56.44	331.25 ± 73.21	341.25 ± 43.03	0.13	0.57	0.93
F/G	1.43 ± 0.08 ^a,b^	1.25 ± 0.05 ^b^	1.17 ± 0.13 ^b^	1.77 ± 0.12 ^a^	1.42 ± 0.07 ^a,b^	1.41 ± 0.06 ^a,b^	<0.05	<0.05	0.67
Serum UN, mmol/L	5.20 ± 0.39 ^b,c^	4.65 ± 0.25 ^c^	4.22 ± 0.16 ^c^	7.26 ± 0.26 ^a^	5.86 ± 0.29 ^b^	5.33 ± 0.16 ^b,c^	<0.05	<0.05	0.16

^1^ RV−, oral gavage of sterile medium; RV+, oral gavage of rotavirus; CON, diet without lactose; 4% Lactose, diet with the addition of 4% lactose; 6% Lactose, diet with the addition of 6% lactose. ^a–c^ Significant difference in values is represented via different letter superscripts in the same row (*p* < 0.05). All data were embodied as means and their SEs (*n* = 6).

**Table 5 animals-12-02336-t005:** Effect of dietary lactose supplementation and/or RV challenge on diarrhea status in weaned pigs ^1^.

Treatments	Sample Number (n)	Normal Number (n)	Diarrhea Number (n)	Diarrhea Rate (%)	*p* Value		
					*P* _4%Lactose_	*P* _6%Lactose_	*P* _RV_
CON	6	6	0	0	0.68	0.07	<0.05
4% Lactose	6	6	0	0			
6% Lactose	6	6	0	0			
CON + RV	6	0	6	100			
4% Lactose + RV	6	1	5	83.33			
6% Lactose + RV	6	5	1	16.67			

^1^ CON, basal diet; 4% Lactose, diet supplemented with 4% lactose; 6% Lactose, diet supplemented with 6% lactose; CON + RV, basal diet with RV challenge; 4% Lactose + RV, diet supplemented with 4% lactose with RV challenge; 6% Lactose + RV, diet supplemented with 6% lactose with RV challenge.

**Table 6 animals-12-02336-t006:** Effect of dietary lactose supplementation and/or RV challenge on rotavirus NSP4 level, antioxidant capacity, and sIgA concentration of jejunal mucosa in weaned pigs ^1^.

	RV−				RV+		*p* Value		
	CON	4% Lactose	6% Lactose	CON	4% Lactose	6% Lactose	Lactose	RV	Lactose × RV
NSP4, µg/mg protein	7.54 ± 0.29 ^c^	7.46 ± 0.25 ^c^	7.40 ± 0.33 ^c^	21.24 ± 0.22 ^a^	10.36 ± 0.46 ^b^	10.03 ± 0.37 ^b^	<0.05	<0.05	<0.05
MDA, nmol/mg protein	0.43 ± 0.02 ^b^	0.42 ± 0.02 ^b^	0.39 ± 0.01 ^b^	0.51 ± 0.01 ^a^	0.48 ± 0.01 ^a^	0.49 ± 0.01 ^a^	0.11	<0.05	0.55
T-AOC, U/mg protein	0.40 ± 0.09	0.43 ± 0.17	0.48 ± 0.16	0.15 ± 0.05	0.22 ± 0.11	0.37 ± 0.07	0.67	<0.05	0.45
sIgA, µg/mg protein	14.34 ± 0.19 ^a^	14.90 ± 0.27 ^a^	15.00 ± 0.14 ^a^	8.58 ± 0.39 ^d^	9.62 ± 0.27 ^c^	10.56 ± 0.25 ^b^	<0.05	<0.05	0.06

^1^ RV−, oral gavage of sterile medium; RV+, oral gavage of rotavirus; CON, diet without lactose; 4% Lactose, diet with the addition of 4% lactose; 6% Lactose, diet with the addition of 6% lactose. ^a–d^ Significant difference in values is represented via different letter superscripts in the same row (*p* < 0.05). All data were embodied as means and their SEs (*n* = 6).

**Table 7 animals-12-02336-t007:** Effect of dietary lactose supplementation and/or RV challenge on serum D-xylose concentration in weaned pigs ^1^.

	RV−			RV+			*p* Value		
	CON	4% Lactose	6% Lactose	CON	4% Lactose	6% Lactose	Lactose	RV	Lactose × RV
Serum D-xylose, mmol/L	0.13 ± 0.06 ^b^	0.20 ± 0.02 ^a,b^	0.35 ± 0.05 ^a^	0.10 ± 0.01 ^b^	0.11 ± 0.04 ^b^	0.23 ± 0.05 ^a,b^	<0.05	<0.05	0.48

^1^ RV−, oral gavage of sterile medium; RV+, oral gavage of rotavirus; CON, diet without lactose; 4% Lactose, diet with the addition of 4% lactose; 6% Lactose, diet with the addition of 6% lactose. ^a,b^ Significant difference in values is represented via different letter superscripts in the same row (*p* < 0.05). All data were embodied as means and their SEs (*n* = 6).

**Table 8 animals-12-02336-t008:** Effect of dietary lactose supplementation and/or RV challenge on morphology of jejunal mucosa in weaned pigs ^1^.

	RV−			RV+			*p* Value		
	CON	4% Lactose	6% Lactose	CON	4% Lactose	6% Lactose	Lactose	RV	Lactose × RV
Villus height, μm	377.40 ± 15.39	375.83 ± 20.80	370.67 ± 38.19	359.25 ± 9.57	371.00 ± 10.46	320.83 ± 8.55	0.38	0.18	0.54
Crypt depth, μm	146.77 ± 7.60	146.98 ± 9.79	138.89 ± 9.21	161.38 ± 14.60	140.00 ± 8.56	129.34 ± 6.49	0.14	0.94	0.41
Villus height/crypt depth	2.55 ± 0.18 ^a,b^	2.58 ± 0.10 ^a,b^	2.91 ± 0.10 ^a^	2.23 ± 0.09 ^b^	2.50 ± 0.09 ^a,b^	2.51 ± 0.13 ^a,b^	<0.05	<0.05	0.38

^1^ RV−, oral gavage of sterile medium; RV+, oral gavage of rotavirus; CON, diet without lactose; 4% Lactose, diet with the addition of 4% lactose; 6% Lactose, diet with the addition of 6% lactose. ^a,b^ Significant difference in values is represented via different letter superscripts in the same row (*p* < 0.05). All data were embodied as means and their SEs (*n* = 6).

**Table 9 animals-12-02336-t009:** Effect of dietary lactose supplementation and/or RV challenge on the mRNA expression of barrier-related and apoptosis-related genes in the jejunal mucosa of weaned pigs ^1^.

	RV−			RV+			*p* Value		
	CON	4% Lactose	6% Lactose	CON	4% Lactose	6% Lactose	Lactose	RV	Lactose × RV
MUC1	1.00 ± 0.01 ^b^	1.06 ± 0.04 ^b^	1.28 ± 0.07 ^a^	0.47 ± 0.05 ^d^	0.51 ± 0.06 ^c,d^	0.71 ± 0. 04 ^c^	<0.05	<0.05	0.90
MUC2	1.00 ± 0.01 ^b^	1.06 ± 0.04 ^b^	1.24 ± 0.06 ^a^	0.21 ± 0.03 ^d^	0.63 ± 0.05 ^c^	0.94 ± 0.04 ^b^	<0.05	<0.05	<0.05
ZO-1	1.00 ± 0.01 ^b,c^	1.56 ± 0.30 ^a,b^	1.88 ± 0.28 ^a^	0.89 ± 0.08 ^c^	1.28 ± 0.06 ^b^	1.63 ± 0.17 ^a,b^	<0.05	0.15	0.91
Occludin	1.00 ± 0.01 ^c^	1.40 ± 0.05 ^a,b^	1.47 ± 0.07 ^a^	0.74 ± 0.06 ^d^	1.27 ± 0.06 ^b^	1.36 ± 0.09 ^a,b^	<0.05	<0.05	0.36
Bax	1.00 ± 0.01 ^c^	1.00 ± 0.01 ^c^	0.87 ± 0.09 ^c^	2.44 ± 0.10 ^a^	1.77 ± 0.07 ^b^	1.53 ± 0.07 ^b^	<0.05	<0.05	<0.05
Bcl-2	1.00 ± 0.01 ^b^	1.17 ± 0.10 ^a^	1.21 ± 0.13 ^a^	0.37 ± 0.05 ^d^	0.58 ± 0.06 ^c^	0.79 ± 0.06 ^c^	<0.05	<0.05	0.35
Caspase-3	1.00 ± 0.01 ^c^	0.93 ± 0.06 ^b,c^	0.82 ± 0.03 ^c^	1.23 ± 0.05 ^a^	1.15 ± 0.05 ^a,b^	1.11 ± 0.03 ^a,b^	<0.05	<0.05	0.58

^1^ RV−, oral gavage of sterile medium; RV+, oral gavage of rotavirus; CON, diet without lactose; 4% Lactose, diet with the addition of 4% lactose; 6% Lactose, diet with the addition of 6% lactose. ^a–d^ Significant difference in values is represented via different letter superscripts in the same row (*p* < 0.05). All data were embodied as means and their SEs (*n* = 6).

**Table 10 animals-12-02336-t010:** Effect of dietary lactose supplementation and/or RV challenge on the gene expression of cytokines and host defense peptides in jejunal mucosa and/or mesenteric lymph nodes of weaned pigs ^1^.

	RV−			RV+			*p* Value		
	CON	4% Lactose	6% Lactose	CON	4% Lactose	6% Lactose	Lactose	RV	Lactose × RV
Jejunal mucosa									
IL-2	1.00 ± 0.02 ^c^	0.98 ± 0.01 ^c^	0.96 ± 0.05 ^c^	5.49 ± 0.15 ^a^	3.83 ± 0.19 ^b^	3.49 ± 0.31 ^b^	<0.05	<0.05	<0.05
IL-4	1.00 ± 0.01 ^b^	1.16 ± 0.06 ^a,b^	1.22 ± 0.06 ^a^	0.83 ± 0.03 ^c^	0.85 ± 0.05 ^c^	0.85 ± 0.03 ^c^	<0.05	<0.05	0.07
IFN-γ	1.00 ± 0.01 ^d^	1.13 ± 0.20 ^d^	1.91 ± 0.28 ^c,d^	2.70 ± 0.20 ^c^	3.17 ± 0.27 ^b^	5.15 ± 0.53 ^a^	<0.05	<0.05	<0.05
IFN-β	1.00 ± 0.01 ^d^	1.15 ± 0.03 ^d^	1.79 ± 0.07 ^c^	2.04 ± 0.05 ^c^	3.29 ± 0.15 ^b^	4.22 ± 0.22 ^a^	<0.05	<0.05	<0.05
pBD1	1.00 ± 0.04 ^c,d^	3.02 ± 0.19 ^b^	4.53 ± 0.56 ^a^	0.48 ± 0.04 ^d^	0.76 ± 0.06 ^c,d^	1.71 ± 0.19 ^c^	<0.05	<0.05	<0.05
pBD2	1.00 ± 0.01 ^c^	2.57 ± 0.37 ^b^	4.54 ± 0.44 ^a^	0.47 ± 0.09 ^c^	1.02 ± 0.06 ^c^	1.13 ± 0.05 ^c^	<0.05	<0.05	<0.05
pBD3	1.00 ± 0.01 ^c,d^	2.44 ± 0.25 ^b^	5.52 ± 0.39 ^a^	0.56 ± 0.07 ^d^	0.96 ± 0.05 ^c,d^	1.62 ± 0.10 ^c^	<0.05	<0.05	<0.05
mesenteric lymph nodes									
IL-2	1.00 ± 0.01 ^c^	0.93 ± 0.01 ^c,d^	0.84 ± 0.02 ^d^	1.40 ± 0.05 ^a^	1.19 ± 0.05 ^b^	1.08 ± 0.10 ^b,c^	<0.05	<0.05	0.26
IL-4	1.00 ± 0.01 ^a^	1.03 ± 0.03 ^a^	1.03 ± 0.05 ^a^	0.55 ± 0.04 ^c^	0.83 ± 0.03 ^b^	0.91 ± 0.05 ^a,b^	<0.05	<0.05	<0.05
IFN-γ	1.00 ± 0.01 ^c^	1.35 ± 0.07 ^b,c^	1.66 ± 0.03 ^b^	2.57 ± 0.05 ^a^	2.79 ± 0.13 ^a^	2.89 ± 0.12 ^a^	<0.05	<0.05	0.11
IFN-β	1.00 ± 0.01 ^d^	1.44 ± 0.06 ^c,d^	1.61 ± 0.12 ^c^	2.26 ± 0.09 ^b^	2.68 ± 0.14 ^a,b^	3.06 ± 0.13 ^a^	<0.05	<0.05	0.55

^1^ RV−, oral gavage of sterile medium; RV+, oral gavage of rotavirus; CON, diet without lactose; 4% Lactose, diet with the addition of 4% lactose; 6% Lactose, diet with the addition of 6% lactose. ^a–d^ Significant difference in values is represented via different letter superscripts in the same row (*p* < 0.05). All data were embodied as means and their SEs (*n* = 6).

**Table 11 animals-12-02336-t011:** Effect of dietary lactose supplementation and/or RV challenge on the number of some bacteria in the cecal digesta of weaned pigs (lg(copies/g)) ^1^.

	RV−			RV+			*p* Value		
	CON	4% Lactose	6% Lactose	CON	4% Lactose	6% Lactose	Lactose	RV	Lactose × RV
*E. coli*	6.97 ± 0.48	6.65 ± 0.40	6.47 ± 0.20	7.35 ± 0.25	7.30 ± 0.31	7.14 ± 0.34	0.58	<0.05	0.90
*Bacillus*	5.93 ± 0.09 ^a,b^	6.07 ± 0.05 ^a,b^	6.12 ± 0.15 ^a^	5.60 ± 0.16 ^b^	5.71 ± 0.17 ^a,b^	5.74 ± 0.21 ^a,b^	0.51	<0.05	0.99
*Bifidobacterium*	4.77 ± 0.30	4.84 ± 0.26	4.93 ± 0.30	4.19 ± 0.1	4.29 ± 0.17	4.66 ± 0.22	0.41	<0.05	0.78
*Lactobacillus*	6.61 ± 0.25	7.26 ± 0.17	7.28 ± 0.29	6.41 ± 0.38	7.01 ± 0.27	7.08 ± 0.23	<0.05	0.33	0.99
Total bacteria	11.21 ± 0.20	11.27 ± 0.17	11.31 ± 0.10	10.87 ± 0.11	11.12 ± 0.21	11.20 ± 0.11	0.39	0.13	0.75

^1^ RV−, oral gavage of sterile medium; RV+, oral gavage of rotavirus; CON, diet without lactose; 4% Lactose, diet with the addition of 4% lactose; 6% Lactose, diet with the addition of 6% lactose. ^a,b^ Significant difference in values is represented via different letter superscripts in the same row (*p* < 0.05). All data were embodied as means and their SEs (*n* = 6).

**Table 12 animals-12-02336-t012:** Effect of dietary lactose supplementation and/or RV challenge on the levels of short-chain fatty acids (SCFA) and pH value in the cecal digesta of weaned pigs ^1^.

	RV−			RV+			*p* Value		
	CON	4% Lactose	6% Lactose	CON	4% Lactose	6% Lactose	Lactose	RV	Lactose × RV
SCFA, µmol/g digesta									
Acetic acid	59.18 ± 5.71 ^b^	63.46 ± 2.60 ^a,b^	70.06 ± 1.61 ^a^	55.01 ± 2.40 ^b^	57.83 ± 2.99 ^b^	63.96 ± 1.70 ^a,b^	<0.05	<0.05	0.95
Propionic acid	29.30 ± 1.94 ^b,c^	31.84 ± 1.92 ^b^	33.99 ± 2.05 ^a,b^	24.78 ± 1.93 ^d^	27.85 ± 1.58 ^c^	35.82 ± 1.48 ^a^	<0.05	0.15	0.19
Isobutyric acid	0.06 ± 0.01 ^a,b^	0.07 ± 0.02 ^a^	0.08 ± 0.01 ^a^	0.03 ± 0.01 ^b^	0.03 ± 0.01 ^b^	0.04 ± 0.01 ^b^	0.59	<0.05	0.96
Butyric acid	15.68 ± 1.49	18.39 ± 2.12	18.40 ± 2.03	13.32 ± 0.57	14.64 ± 1.47	16.27 ± 2.95	0.34	0.10	0.90
Isovaleric acid	0.10 ± 0.01	0.10 ± 0.04	0.10 ± 0.03	0.08 ± 0.01	0.08 ± 0.01	0.10 ± 0.02	0.82	0.40	0.97
Valeric acid	0.53 ± 0.12	0.57 ± 0.08	0.57 ± 0.16	0.34 ± 0.03	0.44 ± 0.05	0.53 ± 0.10	0.60	0.12	0.83
Total SCFA	104.85 ± 2.73 ^b^	114.43 ± 4.38 ^a,b^	123.20 ± 5.31 ^a^	93.56 ± 4.54 ^c^	100.88 ± 4.65 ^b,c^	116.71 ± 5.14 ^a,b^	<0.05	<0.05	0.73
pH value	5.57 ± 0.05 ^a,b^	5.44 ± 0.04 ^b^	5.46 ± 0.13 ^b^	5.84 ± 0.10 ^a^	5.74 ± 0.04 ^a,b^	5.53 ± 0.09 ^a,b^	0.06	0.58	<0.05

^1^ RV−, oral gavage of sterile medium; RV+, oral gavage of rotavirus; CON, diet without lactose; 4% Lactose, diet with the addition of 4% lactose; 6% Lactose, diet with the addition of 6% lactose. ^a–d^ Significant difference in values is represented via different letter superscripts in the same row (*p* < 0.05). All data were embodied as means and their SEs (*n* = 6).

## Data Availability

Not applicable.

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
