# Peer review of "Effect of Dietary Lactose Supplementation on Growth Performance and Intestinal Epithelium Functions in Weaned Pigs Challenged by Rotavirus"

_animals, 2022, doi:10.3390/ani12182336_

Round 1

Reviewer 1 Report

This manuscript (animals-1886083) devoted to the evaluation of the “Effect of dietary lactose supplementation on growth performance and intestinal epithelium functions in weaned pigs challenged by rotavirus” The authors were trying to compare different doses lactose and to find out which doses of lactose could “alleviate RV-induced diarrhea in weaned piglets”.

The article was simple to understand, but the manuscript format is not standardized. There are a lot of writing mistakes in the article, and the language in this manuscript also needs improvement.

Line 25, “as many as possibly”? This sentence is greatly exaggerated. First of all, there is an optimal amount of any nutrient for an animal. In addition, this study is for rotavirus infected pigs. Is it necessary to add a large amount of lactose to healthy piglets?

Line 51 , References need to be checked for formatting errors.

In the introduction, I would suggest the authors first introduce the background of diarrhea caused by rotavirus, and then introduce “based on the physiological function of lactose”, dietary supplementation with lactose may alleviate diarrhea caused by rotavirus. Moreover, there should be add introduction of the harm of rotavirus to newborn piglets and the diarrhea of piglets.

Line 58, Rotavirus is just one of many causes of diarrhea in piglets. What other pathogenic and management factors contribute to piglet diarrhea? What is the relationship between them? Please provide a brief introduction.

Line 65, “Whey has already been used in the feed…”

Line 66, “According to the addition percentage of whey,

Line 66-67, the reference is missing

Line 74, please provide the certificate number of animal care and ethics

Line 79, How old were these pigs? How many male and female piglets? How many litters were they from? Age of weaning?

Line 81, “fed 4 times” per day, How much did you feed each time? How to ensure that all piglets were satisfied? How to record feed intake and wasted feed?

Line 85 and line 109: There is no information of the batch number of lactose and D-xylose solution or where it came from in Methods.

Line 85-86, Were piglets fed pellets or powders? How did you mix a small amount of experimental feed? Equipment? Parameters?

Line 127-129, 137-139, please provide more information (Product name and serial number) about the kit. Elisa kit? accuracy of the kit?

Line 141, Only one sentence for this part. Please briefly introduce the method.

Line 173, “Statistical analysis” 

Please check and correct the grammatical and format errors in the full text.

Table 4, I suggest that Table 4 should be separated into two tables. Because in the RV- column, the pigs at 1-14 d were not the same pigs at 15-20 d. Please use two tables to show these results, one for 1-14 d before RV challenge, one for 15-20 d after RV challenge.

In Table 4, please check the statistical analysis. I don’t believe that the p value of RV for ADG at d 15-20 d is 0.57. Obviously, after RV challenge the ADG decreased in all the lactose treatments.

Table 5, please provide the Chi-square value for each test. I performed Chi-square tests for RV and lactose treatments. But the p value of lactose is not significant in Chi-square test. There is a warning information in SPSS because of a zero value in the Chi-square test for lactose treatment.  Because of the small number of pigs in the test, it is not easy to achieve significant results. It is suggested to statistically analyze the diarrhea days of the piglets. Also, I do not see how to determine diarrhea in piglets in the Materials and Methods.

The titles of the tables should be moved above the tables. Figure captions should be written below them. Please check Table 4 and Figure1, etc.

Line 331-332, reference mistakes in formatting

Line 342-343, “Supplementing lactose in diets improved ADG and F/G of weaned piglets”. However, Table 4 shows p-level for ADG is p=0.09 and 0.13, what means there is no any significant difference among groups. It seems to decrease F/G. Please check this statistical analysis once more.

lines 344-346, references are missing for the sentence

The author suggested that the effect of lactose was dose-dependent, but only two concentrations were set in this study. And there was no linear regression analysis in the statistical analysis.

The article should be written according to the Animals journal format. It seems to be according to the template of Agriculture journal.

Author Response

Reviewer 1

Line 25, “as many as possibly”? This sentence is greatly exaggerated. First of all, there is an optimal amount of any nutrient for an animal. In addition, this study is for rotavirus infected pigs. Is it necessary to add a large amount of lactose to healthy piglets?

Response: Thank you for your suggestion. Since we did not get the result of high dose negative effect in our test addition amount, we cannot explain which addition amount is the highest. We originally wanted to express that we could add more lactose than usual, even more than 6%. However, due to the limitation of cost, the addition amount will not be exaggerated. According to your opinion, we have adopted a more cautious description.

Line 51 References need to be checked for formatting errors.

Response: Thank you for your suggestion. This is indeed our negligence, which has been corrected.

In the introduction, I would suggest the authors first introduce the background of diarrhea caused by rotavirus, and then introduce “based on the physiological function of lactose”, dietary supplementation with lactose may alleviate diarrhea caused by rotavirus. Moreover, there should be add introduction of the harm of rotavirus to newborn piglets and the diarrhea of piglets

Response: Thank you for your suggestion. We have adjusted and supplemented relevant contents according to your opinions.

Line 58, Rotavirus is just one of many causes of diarrhea in piglets. What other pathogenic and management factors contribute to piglet diarrhea? What is the relationship between them? Please provide a brief introduction.

Response: Thank you for your suggestion. There are many pathogens causing piglet diarrhea. Infectious factors include bacteria, viruses, parasites, etc.; Non communicable factors include weaning, malnutrition and poor sanitation. All these factors will more or less cause the intestinal health of piglets to be damaged. The main mechanism of RV causing intestinal injury is invasion of intestinal epithelial cells, leading to inflammatory response, immune dysfunction and oxidative stress. It can be seen that RV will make piglets more susceptible to infection with other pathogens, resulting in the destruction of the overall health of piglets and multifactorial diarrhea.

Line 65, “Whey has already been used in the feed…”

Response: Thank you for your suggestion. We have corrected this paragraph.

Line 66, “According to the addition percentage of whey,

Response: Thank you for your suggestion. We have modified the sentence according to your opinion.

Line 66-67, the reference is missing

Response: Thank you for your suggestion. As the relevant data has not been published, there is no reference, and we have explained it at the end of the sentence.

Line 74, please provide the certificate number of animal care and ethics

Response: Thank you for your suggestion. This is indeed our negligence. We have added the animal care and ethics certificate number and date in the ethical approval.

Line 79, How old were these pigs? How many male and female piglets? How many litters were they from? Age of weaning?

Response: Thank you for your suggestion. We selected 21-day old weaned castrated boars from 12 litters according to the principle of similar weight and birth time. The test was started immediately after weaning, that is, the weaning day was 21 days’ old. We have supplemented relevant contents in materials and methods.

Line 81, “fed 4 times” per day, How much did you feed each time? How to ensure that all piglets were satisfied? How to record feed intake and wasted feed?

Response: Thank you for your suggestion. We have added a description of feeding methods to the materials and methodsThe feeding amount was based on the principle that there was a little remaining feed in the trough after feeding. The residual feed in the trough would be cleaned, weighed and recorded at 10:00 every night.

Our actual feeding operation is consistent with the modified description.

Line 85 and line 109: There is no information of the batch number of lactose and D-xylose solution or where it came from in Methods.

Response: Thank you for your suggestion. We have supplemented the source and purity of lactose and D-xylose. D-xylose was purchased from Chengdu haoboyou Biotechnology Co., Ltd. And the lactose was purchased from Baiying Biotechnology Co., Ltd, Jiangxi, China, with a purity of 99%.

Line 85-86, Were piglets fed pellets or powders? How did you mix a small amount of experimental feed? Equipment? Parameters?

Response: Thank you for your suggestion. We use the model multi-direction motion blender equipment to mix the small amount of experimental powder feed. The equipment was purchased from Jiangsu Yutong drying Engineering Co., Ltd., and the equipment model is SYH-50.

Line 127-129, 137-139, please provide more information (Product name and serial number) about the kit. Elisa kit? accuracy of the kit?

Response: Thank you for your suggestion. We have added the product number, but we are sorry that we have no more information. There are many users of this company's products, and its accuracy is fully guaranteed.

The measuring principle of urea nitrogen is that urea is hydrolyzed under the action of urease to produce ammonia ion and carbon dioxide. Ammonia ion and phenol chromogenic agent in alkaline medium produce blue substance, and the amount of this substance is proportional to the content of urea.

The determination principle of D-xylose is as follows: In strong acid solution, D-xylose is dehydrated to produce furfural, which reacts with phloroglucinol to form a pink compound. The colorimetric determination is carried out at 554nm, and the content of D-xylose can be obtained through calculation.

Line 141, Only one sentence for this part. Please briefly introduce the method.

Response: Thank you for your suggestion. We have supplemented relevant contents.

Line 173, “Statistical analysis” 

Response: Thank you for your suggestion. This is indeed our negligence, which has been corrected.

Table 4, I suggest that Table 4 should be separated into two tables. Because in the RV- column, the pigs at 1-14 d were not the same pigs at 15-20 d. Please use two tables to show these results, one for 1-14 d before RV challenge, one for 15-20 d after RV challenge.

Response: Thank you for your suggestion. After careful consideration, we still think that the existing form is more suitable. If Table 4 is divided into two tables, there will be two tables related to production performance. This will confuse the reader.

In Table 4, please check the statistical analysis. I don’t believe that the p value of RV for ADG at d 15-20 d is 0.57. Obviously, after RV challenge the ADG decreased in all the lactose treatments.

Response: Thank you for your suggestion. We have re conducted the statistical analysis, and the data is indeed true. Our data are presented by the mean ± standard error. From our standard error, we can see that the difference within the group is too large, resulting in the statistical difference is not obvious.

Table 5, please provide the Chi-square value for each test. I performed Chi-square tests for RV and lactose treatments. But the p value of lactose is not significant in Chi-square test. There is a warning information in SPSS because of a zero value in the Chi-square test for lactose treatment.  Because of the small number of pigs in the test, it is not easy to achieve significant results. It is suggested to statistically analyze the diarrhea days of the piglets. Also, I do not see how to determine diarrhea in piglets in the Materials and Methods.

Response: Thank you for your suggestion. We have re conducted the statistical analysis. We modified the table to perform a separate statistical analysis of the two doses of lactose. We found that 4% lactose did not relieve diarrhea, and 6% lactose tended to relieve diarrhea. The chi square values of the two were 0.168 and 3.227, respectively. In order to more conform to the data in the table, we have modified all descriptions of diarrhea. Finally, we added the method of diarrhea score to the trial design and sample collection.

The titles of the tables should be moved above the tables. Figure captions should be written below them. Please check Table 4 and Figure1, etc.

Response: Response: Thank you for your suggestion. We have adjusted the position and format of all tables and their titles.

Line 331-332, reference mistakes in formatting

Response: Thank you for your suggestion. This is indeed our negligence, which has been corrected.

Line 342-343, “Supplementing lactose in diets improved ADG and F/G of weaned piglets”. However, Table 4 shows p-level for ADG is p=0.09 and 0.13, what means there is no any significant difference among groups. It seems to decrease F/G. Please check this statistical analysis once more.

Response: Thank you for your suggestion. We have modified the descriptionThe present research got the similar results that supplementing lactose in diets improved F/G of weaned piglets, and tended to improved ADG of weaned piglets and decrease F/G of RV-infected piglets.

lines 344-346, references are missing for the sentence

Response: Thank you for your suggestion. Our negligence led to this error, which should have been In this study.

The author suggested that the effect of lactose was dose-dependent, but only two concentrations were set in this study. And there was no linear regression analysis in the statistical analysis.

Response: Thank you for your suggestion. We have modified all descriptions of dose-dependent.  

Reviewer 2 Report

The manuscript studied on the effects of lactose supplementation on intestinal health and growth performance of RV infected piglets. It is a novel idea to use lactose to alleviate the damage caused by RV. This is a topic of interest to researchers in related fields. However, the paper needs to be improved before it is accepted for publication.

Introduction: the authors need to discuss the decrease of lactase level in the intestine of piglets and its effect on the amount of lactase entering the hindgut. Lactose equivalents also need to be discussed. The authors provide some background on RV infection, but they must give the reason why they infected animals two weeks after weaning. As the ability of piglets to digest lactose decreases with age, the validity of the conclusion depends mainly on age.

In line 81: 6.30 kg is a weaning weight, please provide the age of the piglets. Which type of diet physical form was used? Please, report.

In line 85 The corn starch and the lactose supplement have the same chemical composition and nutrient levels? The diets, had all the same protein, fat and energy levels when corn starch was replaced with 4 or 6% lactose?

In line 86: the source of lactose and its purity needs to be given.

Table 1: Nutrient content3 instead of Nutrient leve3.

In line 101, "Day 15, all piglets were gavaged", was it RV gavage? Please indicate how the RV was used, when, how often and how many piglets were handled.

Please indicate "n" in in each result.

In line 103: “Scoring method of fecal consistency were carried out as described previously,” however, fecal consistency scores were not shown in the results.

In line 106: correct to: Animals were weighed on days 1, 15 and 21.

In line 118: pH value by pH meter, please provide what type of pH meter.

Is there a literature reference for the amount of lactose added? Are the 4% and 6% concentration gradients too close together to be representative?

The mesenteric lymph nodes appearing in the results section were not collected in the sample collection section.

Author Response

Introduction: the authors need to discuss the decrease of lactase level in the intestine of piglets and its effect on the amount of lactase entering the hindgut. Lactose equivalents also need to be discussed. The authors provide some background on RV infection, but they must give the reason why they infected animals two weeks after weaning. As the ability of piglets to digest lactose decreases with age, the validity of the conclusion depends mainly on age.

Response: Thank you for your suggestion. According to your suggestion, we have supplemented the relevant contents of piglet age and the level of lactase in the introduction. There are two main reasons for RV infection two weeks after weaning. The first point is that at this time, the activity of lactase in the gut will decrease with the increasing age of piglets, this make lots of lactose enter the large intestine. The second point is that diets containing different levels of lactose can reflect the efficacy of lactose after feeding for two weeks.

In line 81: 6.30 kg is a weaning weight, please provide the age of the piglets. Which type of diet physical form was used? Please, report

Response: Thank you for your suggestion. Response: Thank you for your suggestion. We selected 21-day old weaned castrated boars from 12 litters according to the principle of similar weight and birth time. We used powdered feed. We have supplemented relevant contents in materials and methods

In line 85 The corn starch and the lactose supplement have the same chemical composition and nutrient levels? The diets, had all the same protein, fat and energy levels when corn starch was replaced with 4 or 6% lactose?

Response: Thank you for your suggestion. We use lactose to replace corn starch in the same amount. Since both are carbohydrates, we think there is little difference in the levels of fat, protein and energy.

In line 86: the source of lactose and its purity needs to be given.

Response: Thank you for your suggestion. We have supplemented the source and purity of lactose. The lactose was purchased from Baiying Biotechnology Co., Ltd, Jiangxi, China, with a purity of 99%

Table 1: Nutrient content3 instead of Nutrient leve3.

Response: Thank you for your suggestion. We have modified the sentence according to your opinion.

In line 101, "Day 15, all piglets were gavaged", was it RV gavage? Please indicate how the RV was used, when, how often and how many piglets were handled

Response: Thank you for your suggestion. On day 15, all piglets (39-day-old) were administrated 5 mL of sterile 100 mmol/L NaHCO3 solution via oral gavage. After 15 min, half of piglets in each group were gavaged with 5 mL (106 TCID50/mL) of rotavirus (RV +) while the other piglets orally received 5 mL of sterile essential medium (RV-). We have supplemented this content to the manuscript.

Please indicate "n" in in each result.

Response: Thank you for your suggestion. We have added "n" for each table note.

In line 103: “Scoring method of fecal consistency were carried out as described previously,” however, fecal consistency scores were not shown in the results.

Response: Thank you for your suggestion. We have supplemented a brief fecal scoring method after this sentence

In line 106: correct to: Animals were weighed on days 1, 15 and 21.

Response: Thank you for your suggestion. This is indeed our negligence, which has been corrected.

In line 118: pH value by pH meter, please provide what type of pH meter.

Response: Thank you for your suggestion. We used phs-3c acidity meter, and we have added the model to the description.

Is there a literature reference for the amount of lactose added? Are the 4% and 6% concentration gradients too close together to be representative?

Response: Thank you for your suggestion. The main objective of this trial was to study the effect of lactose itself on RV infected piglets. Whey is commonly added in feed production at 5%, and it contains 80% lactose. This was the theoretical basis for 4% lactose addition in the diet of our experiment. The addition amount of 6% has increased by 50% compared with the normal addition amount. And since a higher addition amount will lead to an unbearable cost, we didn’t set a higher addition amount.

The mesenteric lymph nodes appearing in the results section were not collected in the sample collection section.

Response: Thank you for your suggestion. We have supplemented relevant contents in the sample collection

Round 2

Reviewer 1 Report

After the authors’ revision, the manuscript has been improved greatly.

But there are still grammatical problems. I suggest the author to check the full text again before publication.

Line 16-17, “Lactose is a kind of carbohydrate that exists in mammal milk, which has some physiological functions,”

Line 24-26, “we suggest that the diet of weaned piglets should be supplemented with more lactose than 4% lactose in the feed of weaned piglets (especially in the early period) if the cost of feed can be afforded.”

Line 30, “On day 15, half of the piglets were orally infused with RV.”

Line 39-40, please check the parentheses.

Line 42, Please put a full stop at the end of the sentence.

Line 62, “This makes a lot of lactose enter the large intestine”

Author Response

Line 16-17, “Lactose is a kind of carbohydrate that exists in mammal milk, which has some physiological functions,”

Response: Thank you for your suggestion. We have revised this sentence to read:Lactose is a kind of carbohydrate that exists in mammal milks. It has some physiological function.

Line 24-26, “we suggest that the diet of weaned piglets should be supplemented with more lactose than 4% lactose in the feed of weaned piglets (especially in the early period) if the cost of feed can be afforded.”

Response: Thank you for your suggestion. We have revised this sentence to read:Thus, we suggest that the diet of weaned piglets should be supplemented more lactose than 4% lactose (especially in the early period of weaning) if the cost of feeds can be afforded.

Line 30, “On day 15, half of the piglets were orally infused with RV.”

Response: Thank you for your suggestion. We have revised this sentence to read:On day 15, half of piglets in each group were orally infused RV.”

Line 39-40, please check the parentheses.

Response: Thank you for your suggestion. We have adjusted the format of this parentheses.

Line 42, Please put a full stop at the end of the sentence.

Response: Thank you for your suggestion. This is indeed our negligence, which has been corrected.

Line 62, “This makes a lot of lactose enter the large intestine”

Response: Thank you for your suggestion. We have revised this sentence to read: “This increases the level of lactose in the large intestine.”

In addition, we have rechecked the whole manuscript and corrected some grammatical problems. we highlight all the changes with a blue background.
